# Knowledge reuse in software projects: Retrieving software development Q&A posts based on project task similarity

**Glaucia Melo**[1]*, **Toacy Oliveira**[2], **Paulo Alencar**[1], **Donald Cowan**[1]

**1** David R. Cheriton School of Computer Science, University of Waterloo, Waterloo, Ontario, Canada,
**2** System Engineering and Computing Science Program, Federal University of Rio de Janeiro, Rio de Janeiro, Rio de Janeiro, Brazil

* gmelo@uwaterloo.ca

**Data Availability Statement:** All process model and dataset files are available from the https://github.com/glauciams/task2stackRapidMiner.

## Abstract

Software developers need to cope with a massive amount of knowledge throughout the typical life cycle of modern projects. This knowledge includes expertise related to the software development phases (e.g., programming, testing) using a wide variety of methods and tools, including development methodologies (e.g., waterfall, agile), software tools (e.g., Eclipse), programming languages (e.g., Java, SQL), and deployment strategies (e.g., Docker, Jenkins). However, there is no explicit integration of these various types of knowledge with software development projects so that developers can avoid having to search over and over for similar and recurrent solutions to tasks and reuse this knowledge. Specifically, Q&A sites such as Stack Overflow are used by developers to share software development knowledge through posts published in several categories, but there is no link between these posts and the tasks developers perform. In this paper, we present an approach that (i) allows developers to associate project tasks with Stack Overflow posts, and (ii) recommends which Stack Overflow posts might be reused based on task similarity. We analyze an industry dataset, which contains project tasks associated with Stack Overflow posts, looking for the similarity of project tasks that reuse a Stack Overflow post. The approach indicates that when a software developer is performing a task, and this task is similar to another task that has been associated with a post, the same post can be recommended to the developer and possibly reused. We believe that this approach can significantly advance the state of the art of software knowledge reuse by supporting novel knowledge-project associations.

## Introduction

Software development is a knowledge-intensive and collaborative activity [1], facing a constant challenge from the almost continuous change in the underlying technology. Current software development endeavors may combine several programming languages such as Java, Javascript, HTML or SQL with different deployment strategies such as Docker, Kubernetes, AWS, or

**Funding:** MITACS Accelerate. Funding Number: IT11814. GMS. NSERC, Natural Sciences and Engineering Research Council of Canada. PA. COMAP - Centre for Community Mapping. GMS. https://comap.ca/ ELAP - Emerging Leaders in the Americas, Global Affairs Canada. GMS. The funders had no role in study design, data collection and analysis, decision to publish, or preparation of the manuscript.

**Competing interests:** The authors have declared that no competing interests exist.

Google Cloud. As a result, software engineers constantly need to capture and learn new knowledge [2]. During a software development project, which involves working on tasks related to software development, developers must constantly acquire new knowledge and expertise for the project to be successful [2].

Software developers usually use an integrated system of components such as code editors, compilers and debuggers, to build a software product. During this process, developers often need to acquire external support such as code snippets, which can cause the developers to switch frequently between a development environment and knowledge environment, usually implemented through a browser [3, 4]. In other words, developers must leave the development environment, reason about relevant and accurate terms for searches, open a browser, verify the results of the search, check if the source is reliable, and only then, transfer the knowledge obtained to the software product environment [4, 5]. Such activity usually occurs more than once, as software projects are large-scale and iterative. We refer to this effort of tapping into sources of support, reasoning about the help needed and choosing among the vast available content, as **curation**. We use the meaning of curation as inspired by the humanities, which is defined [lexico.com/definition/curate] as:

> "Select, organize, and present material such as online content, merchandise or information, typically based on using professional or expert knowledge."

Essential sources of knowledge, which software developers use for support, are usually question and answer (Q&A) websites. A popular Q&A website used by software engineers is Stack Overflow (SO) [6–9], a community of experts that has become famous as a primary source of knowledge for software developers. Although SO is widely used during software development [6], there are still issues about explicitly associating the tasks performed during development and the knowledge obtained from an SO post [10]. If the curation knowledge could be associated with the tasks, then significant time might be saved, when similar issues are encountered. Searches performed by creating a search string and returning to the Q&A site would be skipped as the solution could be identified and found within the context of the development project.

The lack of integration of the support often needed by software developers with the development project is identified by researchers as an open issue [11, 12], as search results are not related to the context within the software project [10]. Some papers have proposed solutions to integrate SO with the software project through text overlap, but they mainly focus on issues such as bugs and exceptions [12, 13].

Research indicates that developers' expertise largely contributes to the success of software projects [2]. However, current approaches to selecting SO posts that use text overlap do not consider developers' expertise [14]. Additionally, it can be challeng- ing to find the text overlap of SO posts if the project is not managed in English, considering SO content is mainly in English. Moreover, research findings indicate that text overlap between issues and relevant SO content can be as low as 16% [14]. Integrating curated SO posts with software development projects and using similarity between project tasks could help developers avoid redundancy through trying to locate the same solution to a problem multiple times. In addition, integrating SO posts can produce other benefits, such as keeping relevant information in the project, avoiding searches of the same information, helping less experienced developers know how experts are working and reducing workflow interruptions [4].

In this paper, we investigate the possibility of reusing curated SO posts based on project task similarity. We investigate task contexts and submit these contexts to a similarity retrieval model. Precision and accuracy, which are the most common metrics identified among

related works [5, 14–17], were collected. Two research questions guide the evaluation of the implemented model: the first (RQ1) investigates the precision and accuracy of our proposal, facilitating a comparison among other work. The second research question (RQ2) compares different task contexts to understand how different combinations can influence SO post reuse.

Preliminary results have been reported in [18], and the current paper extends previous results in several ways. First, the process model description is extended to support further replication of the proposed model. Second, a systematic mapping study was introduced. This study answers research questions regarding current proposals that associate SO with the development environment. Third, the experimental studies have been extended by the additional distance and similarity algorithm calculations. In addition, the paper has been significantly enhanced by extensions that provide additional details about the background, related work, case studies, and the analysis of the results.

The main contributions of our study are:

- Definition of the term *curation* for the effort developers apply when searching for online support;

- A study of the possibility of capturing and reusing curated Stack Overflow Posts during software development, using existing project task information;

- The detailed demonstration of the correlation between curated Stack Overflow posts and project tasks;

- A Systematic Mapping Study on approaches that associate Stack Overflow with software development. From this study, we have contributed with a detailed updated catalog of the state-of-the-art in the field;

- A data mining implementation, which allowed both development and evaluation of the current study.

This paper is structured as follows. Section Introduction presents an introduction of the subject, followed by Related Works in Section Related Work. Section Systematic Mapping Study presents a systematic mapping study that supports this work and provides important background about the research area. Section Knowledge Reuse in Software Projects presents the study and implementation conducted to investigate the association of SO posts with project tasks. Finally, Section Evaluation evaluates the study and implementation and the conclusions in Section Conclusions, present future work and limitations.

## Related work

A review of the literature was conducted to retrieve approaches that use development artifacts and other information to leverage software development. In summary, some articles suggest development artifacts such as code snippets [5, 19] or API documentation content [15] to search Stack Overflow content. Most articles suggest automated searches for the occurrence of exceptions or issues during the project [14, 16, 17, 19, 20].

In contrast, our approach considers humans-in-the-loop to establish the association between project tasks, described in natural language (i.e., English), and SO posts. For example, a developer performing task1, after searching Stack Overflow, finds out that the post labelled post3 is relevant to task1. The task management system captures (not automatically) the association task1-post3 as developers are performing their tasks. This association is captured as historical data. The activities performed by a developer to associate project tasks with Stack

Overflow posts are named *curation* in our study. When a developer is performing a different task, task2, task2 is similar (i.e., based on a specific similarity metric) to task1, and since there is an historical association (task1, post3), post3 might also be useful for the developer performing task2.

Although successfully used in some applications, existing approaches have some limitations. First, other approaches do not consider humans-in-the-loop to associate a project task with Stack Overflow posts. However, because of their expertise, developers are the primary sources for information related to this association. Secondary sources, such as automated inferences, are not as authoritative as the opinions of developers. Second, because we capture the association as historical data, the link (task, post) is not restricted to a single spoken language. Inter-language associations can be captured when project tasks are described in languages different from Stack Overflow posts. For example, in our study, tasks are captured in the Portuguese language, and the associated SO posts are described in English, as is usual for the vast majority of SO posts. In addition, existing approaches often establish this association based on events. For example, when specific errors (bugs, exceptions) occur, or code is being written, only then does their solution try to find SO posts related to the errors or code. However, our approach does not depend on the occurrence of events and developers having doubts about the tasks they are performing could still find relevant SO posts with the information they need. Developers search for sources of support that can help them solve any problem or concern they might encounter, even if the task they are working on depends on knowledge beyond that which the developers possess [19]. Finally, in our approach, task similarity uses past associations to recommend SO posts. The recommendations are based on the tasks the developers are performing, and do not depend on the similarity with Stack Overflow. The focus of the recommendation is based on the context of the developers' work, and SO information is not needed. In contrast, other approaches focus on finding similarities between the project and Stack Overflow directly, and this association, as mentioned previously, is focused on specific topics, such as bugs, code and issues. The combination of having similarities that focus on task context and are based on historical data is a step forward in the state-of-the-art of associating software development project tasks with knowledge repositories, such as Stack Overflow.

Regarding the use of historical data, this subject has been explored by some authors to find and recommend artifacts [21]. The work of ČubraniĆ et al. [22, 23] uses artifacts created within the project, which is similar to what is proposed in this paper, and records what is called "project memory." The "project memory" is a logical association of artifacts, providing developers with a source of associations of recurrent use of artifacts. This approach proposes that a high number of artifact associations is useful to suggest to developers which artifacts they should then access, in case the artifact matches the "project memory."

More recent papers investigate approaches for developers that build upon the current power of Stack Overflow and other software development question-and-answer systems (Q&As). Jing Li et al. [24] conduct exploratory and hypothesis validation studies that indicate that the presence of hyperlinks in Stack Overflow posts have the potential to aggregate information for developers in a number of other web resources such as official APIs, tutorials, code examples and forum discussions. Jing and Sun Li et al. [25, 26] discuss the challenges and strategies for facilitating and promoting answers to developers' questions in software documentation through the content of Stack Overflow posts. They use the context in SO posts to identify SO question-documentation pairs incorporating the content of software documentation and social context on Stack Overflow into a learning-to-rank schema. A solution to distill API negative red flags from SO automatically is also proposed in [27].

## Systematic mapping study

In this section, we describe the method used to conduct the systematic mapping study using the approach suggested by Petersen et al. [28], towards providing a mapping for work that associates software development with Stack Overflow.

The Systematic Mapping Study (SMS) approach is a common method to obtain information on a particular subject and provides categorized results that have been published [28, 29]. The SMS reported in this paper was conducted to gather state-of-the-art research in the literature, related to current solutions that associate SO with software development. This section presents the protocol used to select studies for this research. This protocol is the process of building search strings and defining a search scope. These procedures were used to provide a comprehensive examination of what has been published on the specific topic of strategies to associate development project tasks with SO posts, acknowledging the current proposals, the input and output information, the evaluation method and the results of each paper. Regarding the search scope, we wanted to select database repositories that (i) contained relevant repositories, such as ACM and IEEE and (ii) allowed searches based on a combination of Title-Abstract-Keywords tokens.

The suggested protocol [28] employs the goal, question, metric (GQM) approach [30] to define goals for the SMS. The goal of this SMS is to **analyze proposals** that associate software development with SO **with the purpose of** characterizing these proposals **regarding** strategy, input, output, evaluation methods, metrics, and results **from the point of view of** researchers **in the context of** software development.

The search described in this paper has as an objective the analysis of papers whose purpose is associating software development with SO. It is possible to replicate this SMS, considering how it was organized into plan, execution and analysis steps. This type organization also provides a summary of the papers that connects SO with software development. Since the objectives of this SMS are defined, the next sub-sections present the plan and execution steps of the SMS.

### Planning

The planning includes: (i) defining the research questions according to the objectives, (ii) using a Problem, Intervention, Comparison and Outcome (PICO) [31] approach to aid in the construction of the search strings and finally, (iii) creating a search string. According to the objective for this SMS, given that this research refers to proposals that aim to associate software development with SO, the research questions for this SMS are:

**RQ1:** *What are the existing association strategies?* It is important to identify how researchers attempt to solve the association between software development and SO, mapping their strategies and approaches, how they developed their solutions and what are the differences and common aspects of each solution.

**RQ2:** *What are the input and output information?* Software development has different contexts that can be used as input to search for SO information such as code, exceptions, tags and project tasks. At the same time, SO can retrieve different types of content such as posts, tags, user information, code chunks, comments, and questions or answers.

**RQ3:** *What are the evaluation methods?* Understanding the methodology used to evaluate the proposals is important, as researchers can use the same source of data to compare approaches, or even position their own dataset in regard to datasets other researchers have used.

**RQ4:** *What are the reported metrics and results?* Explicitly demonstrating metrics and results of each implemented approach, is besides a summary, a procedure to inform researchers on the current used metrics and methods.

To organize and structure the search string based on the objective and research questions, the PICO approach was used [31]. Table 1 presents a PICO description and goals.

According to the PICO definitions, the search string (and synonyms) is:

"stack overflow" OR "stackoverflow"

AND (issue OR task OR context)

AND (recommend* OR suggest* OR associat* OR link OR integrat*)

AND ("software"OR "software process"OR "develop*"

The scientific databases chosen were: Scopus, Web of Science, IEEE Xplore and ACM.

For the selection of articles, Inclusion and Exclusion criteria were established. These criteria support the decision about whether articles should be read or ignored. The selection criteria are outlined next.

- Inclusion Criteria:

  - Articles related to SO or communities where developers search for support; OR

  - Articles related to the association or recommendation or suggestion or link of software project artifacts; OR

  - Articles that address the use of software development information to find SO posts;

- Exclusion Criteria:

  - Articles not in the computer science area; OR

  - Articles whose research does not focus on Software Engineering; OR

  - Proposals that are not applied to software development; OR

  - Articles not written in the English language; OR

  - Articles that do not state clearly the association strategy, input, output, evaluation method and results between SO posts and software development; OR

  - Articles published before 2010.

The group of control articles [28] that should be retrieved by the search string is:

[10] PONZANELLI, L. et al. Mining StackOverflow to Turn the IDE into a Self-Confident Programming Prompter. Proceedings of the 11th Working Conference on mining software repositories, p. 102-111, May 31, 2014.

**Table 1. PICO [31] for SMS.**

| (P) Population | Software Development |
|---|---|
| (I) Intervention | Research that aims at associating development projects with Stack Overflow |
| (C) Comparison | Not applicable, as the goal of this study is to identify the state-of-the-art, not to compare with other work |
| (O) Outcomes | Solutions that associate software development with Stack Overflow |

[12] PONZANELLI, L.; BACCHELLI, A.; LANZA, M. Seahawk: Stack Overflow in the IDE. Proceedings of the 2013 International Conference on software engineering, p. 1295-1298, May 18, 2013.

[20] CAMPOS, E.C.; SOUZA, L.B.L.; MAIA, M.D.A. Searching Crowd Knowledge to Recommend Solutions for API Usage Tasks. Journal of Software: Evolution and Process, Chichester, v. 28, n. 10, p. 863-892, Oct 2016.

[14] CORREA, D.; SUREKA, A. Integrating Issue Tracking Systems with Community-Based Question and Answering Websites. 2013 22nd Australian Software Engineering Conference, p. 88-96, 2013.

The papers were selected according to the following three steps:

- **Step 1—Preliminary selection of publications**: Execute the search string on databases to perform the preliminary retrieval of papers and use a reference manager to store retrievals;

- **Step 2—Selection of relevant publications—1st filter**: For an initial selection of relevant papers, the title and abstracts of each returned paper are read and assessed according to the inclusion and exclusion criteria. For some papers, reading only abstracts may not be sufficient to determine whether or not they should be included. For this reason, the papers are selected for full reading (step 3). Also, for databases that return many results (+200), we have ordered results by relevancy (relevancy selection) and applied abstract and title reading. After 2 pages with no new findings according to inclusion and exclusion criteria after title and abstract reading, we have stopped the search;

- **Step 3—Selection of relevant publications—2nd filter**: After removing duplicates, all papers selected in the 1st filter (step 2) will be fully read and analyzed to determine if they match the inclusion and exclusion criteria set. The selection of papers is final.

## Execution of the search

According to the procedures for selecting papers defined in the planning of this SMS, the next step is to perform the execution of the search string in each of the selected sources. Table 2 presents the total number of papers retrieved from each of the databases.

We have eliminated articles based on titles and abstracts (step 2), and full-text reading (step 3). Additionally, we have decided to add articles using backward snowball, as the retrieval of articles was conducted by the first author and this may pose a threat to the validity of the study (see further discussion in Section Threats to Validity). We have also decided to include papers for full-text reading when in doubt, also to mitigate the possible threats for this study. The second author has reviewed the planning and execution criteria of the SMS and search string testing.

To complement the automated search query, we have conducted a manual search to find articles that were not retrieved by the search query. Our search string did not locate one important source, namely the paper from Čubranić et al. [22]. Since this paper is well-

**Table 2. Number of papers retrieved from each database.**

| Database | Search results |
|---|---|
| Scopus | 127 |
| Web of Science | 442 |
| IEEE Xplore | 410 |
| ACM | 46 |

referenced in the literature, we included this publication in our list of research results through manual search. As a result of the manual search, to the best of our knowledge, we found that only this article should be included because it also addresses the general problem of using historical data to find software development artifacts. This paper was also identified by the snowballing process, but not included, as we were already considering this paper inclusion from manual search (identified as a duplicate study).

All articles retrieved by each of the selected databases (Step 1) were stored in a reference manager tool and for databases that returned many results, relevancy selection was executed (Step 2) before storing the articles. Duplicated articles were removed, resulting in 163 articles that had the title and abstract read (Step 2), and when applying inclusion and exclusion criteria, 50 articles resulted. After fully reading these articles (Step 3), twenty-five articles were finally selected, one article added through the manual technique. From snowballing, three more articles were added. These final 29 articles are presented in Table 4. Fig 1 presents the selection/elimination process with the number of articles of each step of the process in detail. In Table 3, we have created an identifier for the papers (ID) used to reference each paper.

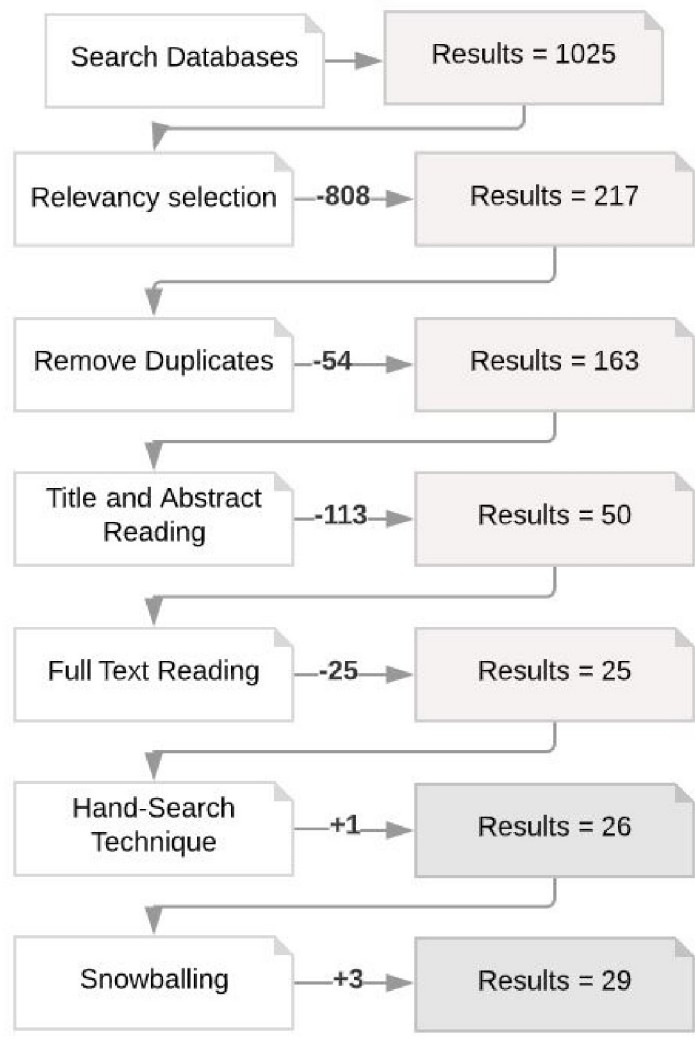

**Fig 1. Number of papers in each sub-step of selection process.**

**Table 3. Attributes extracted from selected papers.**

| Attributes | RQ | Description |
|---|---|---|
| ID | - | Identifier of study (to match Table 6) |
| Association Strategy | RQ1 | The strategy mapped in each study used to associate software development with Stack Overflow |
| Input | RQ2 | What was the input information used in each study to generate the association between software development and Stack Overflow? |
| Output | RQ2 | What was the output information generated by the association strategy? |
| Evaluation | RQ3 | Sources and characteristics of the datasets used to evaluate proposals |
| Results | RQ4 | The results of the evaluation |

After reading each of the 29 retrieved papers, the attributes extracted are presented in Table 3. The attributes were created based on the research questions. Each field has an item as well as a description of the attribute. The attributes presented in Table 3 were created based on the SMS research questions. All articles presented in Table 4 after being fully read had their attributes (Table 3) extracted and presented in Table 1 in Section S1 Appendix. Systematic Mapping Study Resulting Catalog. The last row of this table presents the results of this proposal, which were later added to the table for comparison purposes, but the Systematic Mapping Study was conducted before the final implementation of the current study. The analysis and discussion of the extracted data are presented in Section Analysis. Note that some of the papers refer to the same study, therefore, they are listed in Table 4 and the reference study is presented in Table 1 in S1 Appendix. The last column of Table 1 in S1 Appendix. refers to the current proposal presented in this paper. These results were later added to this table for comparison purposes, but the Systematic Mapping Study was conducted before the final implementation of the current study.

During the snowballing, 22 new articles were retrieved from analyzing the chosen papers title's references. After reading title and abstract, 9 papers were chosen to be fully read, and we ended up adding another 3 papers to the final selection. The papers are identified in Table 4 and Table 1 in S1 Appendix. by the sufix *SB*.

## Analysis

The analysis of the information extracted from the papers gathered during the procedure is presented in this section. For this paper, a thematic rationale [28] to discuss the details of the papers is proposed. The authors of this paper were responsible for extracting the information from each retrieved paper and based on the information extracted from the papers, it was possible to answer the research questions of this SMS. Each research question is referenced in Table 1 in S1 Appendix and the next sub-sections provide further detail on each of these research questions.

**RQ1: What are the existing association strategies?** Researchers have attempted to evaluate the impact of defining a context as well as using text similarity to find correlations between them to recommend artifacts from the project history, such as Hipikat (S1). Hipikat (S1) does not use SO information, it uses other artifacts that have been created during the project such as other tasks, source file versions, messages on forums and project documents. There is no external associated context such as an SO post. In other words, S1 does inferences of links by combining information contained within the project artifacts and the meta-information about these artifacts from different information sources. Some of the authors of this paper (S1) have worked on Mylar [50], which is a well-known relationship heuristic for software development

**Table 4. Systematic mapping study selected articles after Step 3.**

| ID | Year | Authors | Title |
|---|---|---|---|
| S1 | 2006 | ČubraniĆ, Davor, Gail C. Murphy, Janice Singer, and Kellogg S. Booth [22] | Learning from project history: a case study for software development (Hipikat) |
| S2 | 2018 | Greco C., Haden T., Damevski K. [19] | StackInTheFlow: Behavior-driven recommendation system for stack overflow posts |
| S3 | 2018 | Wu D., Jing X.-Y., Chen H., Zhu X., Zhang H., Zuo M., Zi L., Zhu C. [15] | Automatically answering API-related questions |
| S4 | 2016 | Campos E.C., de Souza L.B.L., Maia M.D.A. [32] | Searching crowd knowledge to recommend solutions for API usage tasks |
| S5 | 2015 | Wang T., Yin G., Wang H., Yang C., Zou P. [16] | Automatic knowledge sharing across communities: A case study on android issue tracker & stack overflow |
| S6 | 2014 | Rahman M.M., Yeasmin S., Roy C.K. [13] | Towards a context-aware IDE-based meta search engine for recommendation about programming errors and exceptions (SurfClipse) |
| S7 | 2013 | Correa D., Sureka A. [14] | Integrating issue tracking systems with community-based question and answering websites |
| S8 | 2013 | Ponzanelli L., Bacchelli A., Lanza M. [5] | Leveraging crowd knowledge for software comprehension and development (Seahawk) |
| S9 | 2012 | Bacchelli A., Ponzanelli L., Lanza M. [33] | Harnessing Stack Overflow for the IDE (Seahawk) |
| S10 | 2015 | Wang, W., Malik, H., Godfrey, M.W [34] | Recommending Posts Concerning API Issues in Developer Q&A Sites |
| S11 | 2015 | Amintabar, V., Heydarnoori, A., Ghafari, M. [35] | ExceptionTracer: A Solution Recommender for Exceptions in an Integrated Development Environment |
| S12 | 2014 | Ponzanelli, L., Bavota, G., Di Penta, M., Oliveto, R., Lanza, M [36] | Prompter: A Self-confident Recommender System |
| S13 | 2017 | Rahman, M. M., Roy, C. K., Lo, D. [37] | RACK: Code Search in the IDE using Crowdsourced Knowledge |
| S14 | 2016 | Sahu, T. P., Nagwani, N. K., Verma, S. [38] | An Empirical Analysis on Reducing Open Source Software Development Tasks using Stack Overflow |
| S15 | 2016 | Fumin, S., Xu, W., Hailong, S., Xudong, L. [7] | Recommendflow: Use Topic Model to Automatically Recommend Stack Overflow Q&A in IDE |
| S16 | 2016 | Liu,X., Shen,B., Zhong,H., Zhu,J. [8] | EXPSOL: Recommending Online Threads for Exception-related Bug Reports |
| S17 | 2018 | Wang,H., Wang,T., Yin,G., Yang,C. [39] | Linking Issue Tracker with Q&A Sites for Knowledge Sharing across Communities |
| S18 | 2018 | Zhang, F., Niu, H., Keivanloo, I., Zou, Y. [40] | Expanding Queries for Code Search Using Semantically Related API Class-names |
| S19 | 2018 | Melo, G., Oliveira, T., Telemaco, U., Alencar, P., Cowan, D. [18] | Towards using task similarity to recommend Stack Overflow posts |
| S20 | 2019 | Rahman, M.M. [41] | Supporting Code Search with Context-Aware, Analytics-Driven, Effective Query Reformulation |
| S21 | 2019 | Cai, L., Wang, H., Huang, Q., Xia, X., Xing, Z., Lo, D. [42] | BIKER: A Tool for Bi-Information Source Based API Method Recommendation |
| S22 | 2019 | Al-Batlaa, A., Abdullah-Al-Wadud, M., Anwar, M. [43] | A Method to Suggest Solutions for Software Bugs |
| S23 | 2019 | Rahman, M. M., Roy, C. K., Lo, D. [44] | Automatic query reformulation for code search using crowdsourced knowledge |
| S24 | 2019 | Yin, H., Sun, Z., Sun, Y., Jiao, W [45] | A Question-Driven Source Code Recommendation Service Based on Stack Overflow |
| S25 | 2020 | Uddin, G., Khomh, F., Roy, C. K. [46] | Mining API usage scenarios from stack overflow |
| S26 | 2019 | Silva, R., Roy, C., Rahman, M., Schneider, K., Paixao, K., Maia, M. [47] | Recommending Comprehensive Solutions for Programming Tasks by Mining Crowd Knowledge |
| S27SB | 2010 | Brandt, J., Dontcheva, M., Weskamp, M., Klemmer, S. R. [48] | Example-Centric Programming: Integrating Web Search into the Development Environment |
| S28SB | 2012 | Cordeiro, J., Antunes, B., Gomes, P. [49] | Context-based recommendation to support problem solving in software development |
| S29SB | 2014 | de Souza, L. B., Campos, E. C., Maia, M. D. A. [20] | Ranking Crowd Knowledge to Assist Software Development |

artifacts. All other papers use text similarity to match the development information with SO, except S5 and S6. S5 uses strategies other than text association to increase the accuracy of results, such as Temporal Similarity and the discovery of Internal Links of SO in the issue's text body. SurfClipse (S6), for example, uses a context-aware tool to identify possible search queries for exceptions presented in the integrated development environment (IDE). The strategy is to search for similar texts presented in the exception and another context in the IDE.

Through the identified context, a SO corpus is built containing the exception and the code that should help the developer fix the exception. S11 and S28 also propose solutions to aid development when an exception occurs. A few works propose solutions to support software engineers when developing code for APIs (S3, S10, S13, S18, S25, S26). They differ in their API coverage, i.e., S26 extends S21 because it not only suggests a limited set of APIs.

**RQ2: What are the input and output?**   Seahawk (S8) uses pieces of the code written by developers in Eclipse to search for entire code snippets, saving developers the time they spend in typing an entire block of code that is already in SO. Other works that use pieces of code as input are S9, S12 and S15. S27 uses code too, but it also augments with context such as the programming language and framework versions. The approach used in Crosslink (S5) is different, as it uses text similarity to match Android Issues with SO issues, recommending the most similar posts. Similarly, the work from Correia et al. (S7) also uses the text related to the issue to associate SO with Android Issues. This work also analyzes contextual features (such as question tags representing the topic) to recommend SO questions in response to bug reports. S17, S19 and S26 also use as input issues or text from tasks, and S25 combines code and task description. The approach proposed by Souza et al. (S4) classifies SO pairs of question/ answers, comparing these pairs with the developer's code through text similarity, and also focuses on retrieving code snippets. The work from Wu et al. (S3) associates' text from API Tutorials with SO posts, using Cosine Similarity. It retrieves questions regarding the API on which the developer is currently working. StackInTheFlow (S2), a recent work published in 2018, similarly to the aforementioned, is a tool integrated with the IDE that uses the source code to generate SO queries. It uses information and events occurring in the environment to generate the queries and present the results to developers. For example, these events can be the occurrence of an error. The tool automatically searches Stack Overflow with the text of the error that occurred. The tool also analyzes when developers are having difficulties in coding, these difficulties being revealed through deleting a wide range of code or not typing anything for a certain amount of time. It starts a search using the code the user is currently editing. Users can also use this tool to search for SO posts manually.

All selected papers have as output an SO post or a code snippet that is part of an SO post. Hipikat (S1) is the exception, as it does not recommend SO posts, but it recommends other artifacts from the software project, such as project documents and source file versions. Other papers deliver minimally an SO post or a code snippet from a SO post or the web to developers.

**RQ3: What are the evaluation methods?**   When considering characteristics of the evaluations of the selected papers, works perform both qualitative and quantitative evaluations (some works report both). Still, quantitative evaluations are more prevalent among the analyzed papers. Recommendation methods struggle with the cold-start problem. Other than that, when considering software engineering, there is a need to engage in studies that take time and could depend on industry collaboration. Most of the selected papers use relatively small samples to validate their studies.

S1, S2, S19, S25 and S27 provide qualitative evaluations, while the remaining papers offer a quantitative evaluations. S1 uses an evaluation method that compares how easy it was for newcomers in a company to get artifact relations using Hipikat and compared those results with tasks executed by experienced developers. S2 analyzed the log of the tool's usage and considered the high number of clicks as a success. The other papers that offered a quantitative evaluation, used samples with the following number of items in the samples: S3 used 5 APIs and 30 SO posts for each; S4 used 35 questions from cookbooks from 3 different technologies, with 12 for the first, 14 for the second and 9 for the third cookbook. S5 used a large number of issues from the Android Issue Tracker. Android Issue Tracker, at the time of the collection of the

dataset, had a total of 151,815 issues and 30,572 threads, and a total of 653 direct links to SO. The time period reported is between November 2007 and September 2013. S6 uses 75 traces from the IDE, with 38 stack traces from logs of 6 grad students and 37 common exceptions from Java traces. S7 uses two samples from platforms such as Google Chromium and Android Issue Tracker. Both of these platforms contain issues that software developers around the world have inserted. They matched the explicit links to SO encountered in both platforms to the actual text of the Issue. The authors also surveyed software maintenance professionals to investigate solutions to common problems in the area. By the time this paper was published, the final results have not been collected. S8 uses a sample of three experiments with 35 Java exercises from books. S28 collects thirty questions from SO and queries these posts using defined context and keywords. S16 extract pairs of issues on GitHub and SO and tries to identify semantic similarities between bug fixes and corresponding SO threads. S17 clusters posts based on the semantic similarities and components diameters, and sets a similarity threshold of 30%. As seen, evaluation methods differ a lot depending on the proposal and hypothesis authors want to evaluate.

**RQ4: What are the reported metrics and results?**   Among the papers that have reported quantitative metrics, precision is reported as the most common metric, reported in S3, S5, S7, S16, S17, S18 and S27. Mean Reciprocal Rank (MRR) was reported by S3, S5, S16, S21, S22 and S25. Normalized Discounted Cumulative Gain (NDCG) was reported by S4 and S8. Accuracy was the metric used by S6, S8, S10, S24 and S26. S2 presents results for different types of evaluations that were performed. The types are: manual, difficulty, user action and error. These types were the actions that triggered the generation of the queries. Manual is when the developer manually creates a query and difficulty is when the system identifies through some parameter that the developer is having difficulties while programming. User action is when the user performs one of the set of actions expected by the tool. Finally, an error is when an exception is raised.

## Threats to validity

We have identified as internal threats in this systematic mapping study that human performance might affect the selection of papers. To mitigate this threat, we used protocols and systematic mapping study established guidelines [28] to enable search replication and minimize bias. As external threats, we identified the use of few research databases can lead to missing studies; we resolve this threat by using four important databases in the Computer Science field. For conclusion threats, we identified that the interpretation of data (selected papers) is a concern once bias can be introduced and only one researcher chooses the papers. The search was executed three times (January 2018, September 2018 and May 2020) aiming at mitigating this threat. Additionally, we have implemented snowballing and the inclusion of papers for full-reading when in doubt and the second author has reviewed in detail the protocol planning and execution of the SMS, both being guidelines suggested by established literature [28].

## Knowledge reuse in software projects

Knowledge can be defined as a "fluid mix of framed experience, values, contextual information, expert insight and grounded intuition that provides an environment and framework for evaluating and incorporating new experiences and information. It originates and is applied in the minds of **knowers**. In organizations it often becomes embedded not only in documents and repositories but also in organizational routines, processes, practices and norms" [51, 52].

As software development occurs, developers deal with growing amounts of knowledge, and in the perspective of the definition given above, developers are the *knowers*. This knowledge is

a valuable asset when stored and managed and, therefore, reused. [53]. Researchers have stated that knowledge in software development can be explicit, in the form of a code and artifacts. Furthermore, knowledge reuse is more than code reuse. As stated by Barnes and Bollinger [54], "The defining characteristics of good reuse is not the reuse of software *per se*, but the reuse of human problem-solving". Among the significant kinds of knowledge that can be reused in software development, tacit knowledge is one of them [55]. Recent research claims there is a current trend for software reuse evolving towards knowledge reuse, as purely reusing software is not easy for developers, nor profitable or interesting [56, 57]. In this work, we consider knowledge to be the information gathered from sources of support, integrated with semantic content tailored by software developers' expertise. In this case, Stack Overflow posts integrated to the software project.

Associating SO posts previously used to support developers with project tasks should allow reusing SO posts retrieved by the tacit knowledge activity of SO post curation. Similar project tasks should be discovered by using a discovery engine that can compute project task similarities. In this section, we present (i) a study overview, an investigation on project task context elements, and an implemented model that extracts similarities between project task context elements. Finally, Section Evaluation presents an evaluation of the proposed model.

## Overview

Curation in this situation is the act of searching and selecting useful SO posts for a problem encountered during a software development project. The process of curating SO posts is described by the steps enumerated later in this section. Each of these steps can be executed repeatedly until developers are satisfied with the results listed and have chosen an SO post that meets their needs. Once a solution is chosen, curation is over. If results are not helpful, the developer changes the string, trying to use new terms that might provide useful results. The curation steps are represented as a Business Process Model and Notation (BPMN) model in Fig 2.

1. Create search string: A developer has a problem and creates a search string that may retrieve satisfactory results;

2. Submit search string: The search string is submitted to SO;

3. Search and list results: SO executes the search according to internal algorithms and lists the results;

4. Select results: The developer selects one (or a set of) SO post that might be helpful.

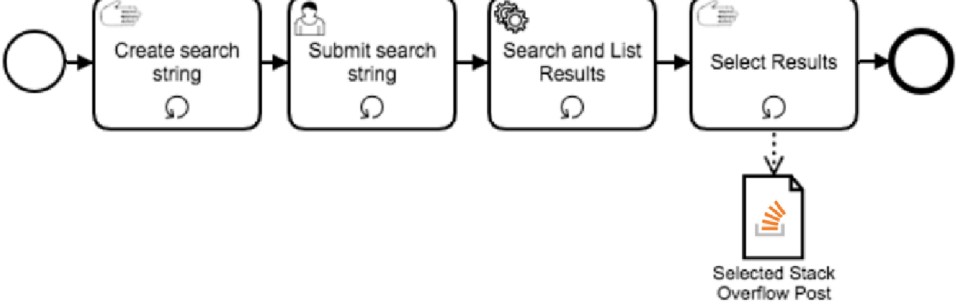

**Fig 2. BPMN model representing the curation process.**

Frequently, after a solution from an SO post is selected, the post containing the solution is not associated with the project task and, therefore, cannot be reused by developers dealing with similar or recurring tasks. A growing body of literature recognizes the importance of associating external knowledge (information developers look for on the web) with the related software development task [10–12, 14, 58]. SO is the source of support considered in this paper, and this support is presented in the form of curated SO posts. Through the identification of similar project tasks, SO posts that were once used to support a project task during its solution could be automatically associated with project tasks with similar contexts. Through the identification of similar project tasks, SO posts that were once used to support a project task during its solution could be automatically associated with project tasks with similar contexts.

As for the mechanism that will allow reuse, we propose to find similar project tasks, in regards to their context, and reuse the same SO post on similar new tasks. Our study aims at using project task information and a text similarity to aid SO Post curation and reuse during software development. SO posts that were once used to support a task during the task solution could be automatically linked to a similar task that is encountered. Fig 3 illustrates this process. The processing functionality receives project task contexts as input, submitting these tasks to a similarity retrieval engine and, by discovering the similarity between pairs of tasks, links the SO Post of a project task to the most similar project task identified by the similarity retrieval process.

In Fig 3, there are four tasks in the Project (1). Each project task is composed of information, which we refer to as "Context" (2). Three of these tasks (task1, task2 and task3) have a status of "resolved", represented by the white color of the "Task Status" caption (3), and have SO Posts (4) associated with them. One task is not resolved (task4), nor does this task have an associated SO Post. When all the tasks are submitted to a similarity retrieval algorithm (5), the similarities between the project tasks are retrieved (6). These similarities are expressed in a range of 0 to 1. In the example, when comparing task4 to task1, the similarity index between them is 0.6, meaning these tasks are 60% similar. When comparing task4 to task2, the retrieved similarity is 0.7, meaning these tasks are 70% similar. Finally, when comparing task4 to task3, the result produced was a 30% similarity between these two tasks. Based on the proposed approach, the most similar tasks should share the same SO Post (7). Thus, task4 should be related to the SO Post associated with task1 and task2, as they are the tasks most similar to task4 based on the similarity index used in the illustration.

Developers perform the work of associating SO posts to project tasks (4), after they have curated an SO post. Therefore, part of the solution is a developer's responsibility. The

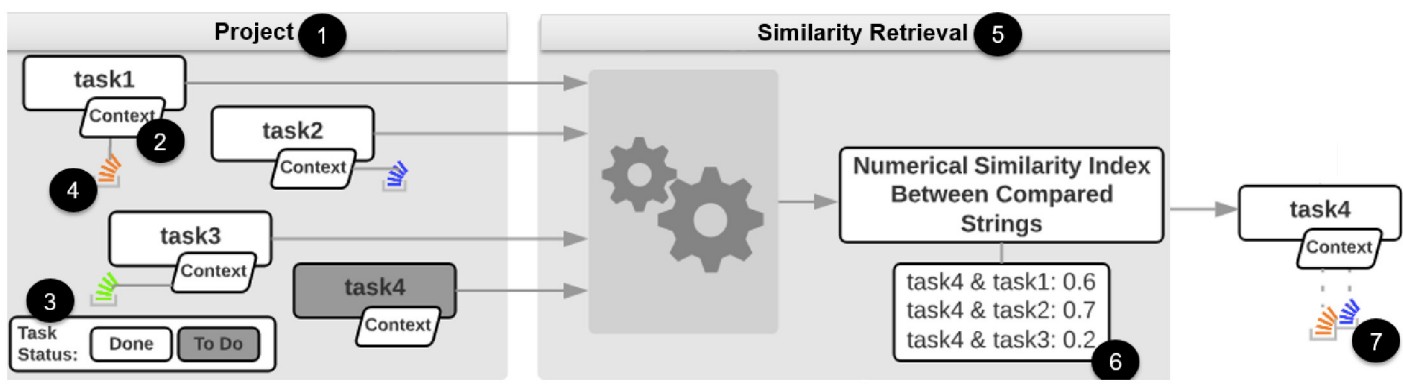

**Fig 3. Proposed similarity approach overview.**

suggested associations created automatically by the similarity retrieval execution are represented in Fig 3 by marker 7. This proposed approach does not consider a direct association of the project tasks with SO. The curation effort is the process that associates the project task to the curated SO post, highlighting the need for developer to be involved in the curation effort.

Developers perform the work of associating Stack Overflow posts to project tasks (4), after they have curated this SO post. Therefore, part of the solution is one developer's responsibility. The association's suggestions created automatically by the similarity retrieval execution are represented in Fig 3 by marker 7. This proposed approach does not consider a direct association of the project tasks with Stack Overflow. The curation effort is what associates the project task to the curated Stack Overflow post, highlighting the need for developer effort for curation.

After a preliminary investigation [18], this enquiry was developed further. In this preliminary investigation, we have performed a qualitative evaluation, asking software developers if similar tasks found among 4000 project tasks could reuse the same Stack Overflow posts. Results c 30% of the possibility of reuse pointed out by the developers that participated in the study. As of the current research, we investigated the elements of the context of a project task and outstanding results are presented in Section Project Task Context. Then, we implemented a process using this project-task context using a data science platform (RapidMiner —rapidminer.com). The results of this research are in Section Similarity Model Implementation. Section Evaluation evaluates and further discusses the similarities retrieved by this implementation.

## Current and foreseen situations

This Section provides details about the current situation related to associating SO posts with software development tasks and the key additions provided by the research described in this paper. Currently, developers search for an SO post, use the information found (if useful), and go on to the next task in the project. The developers do not record the SO post, how they found it, or the work they had to employ to transform the information obtained on SO into useful information for the project.

We argue that the knowledge supplied by the SO post could be valuable in the project in future, but because the developer does not associate the information of the useful SO post with the project, this knowledge is lost. If the same developer or another member of the team is working on a similar task, this person will have to look for the same SO post again, and the curation effort is duplicated. Our proposal aims to reuse the curation required from the first developer. This first search for SO posts (curation) will have to be executed. But once executed, and associated with the project task that required the extra support, the curation effort can be minimized or even avoided.

There are still two aspects of reuse that have to be considered: the association of the SO post with the project task and the mechanism that will allow the reuse of the curation. First, the association is solved by the first developer, who performed the curation, manually providing information to the tool that manages the project tasks and indicating, which SO post supported that specific task. Recording such information would require tool customization. For example, this association can be performed by creating a customized field used by the project management tool to store SO posts used. To illustrate Fig 4 shows an example of a project task of the Redmine open source project. This project task has as context: a title, description and other information regarding the project task. If the developer that solves this task needs SO support, the developer could describe which SO post was used in a customized field, as presented in Fig 5. Note that this field does not exist in Redmine, we are proposing this illustration for clarification purposes.

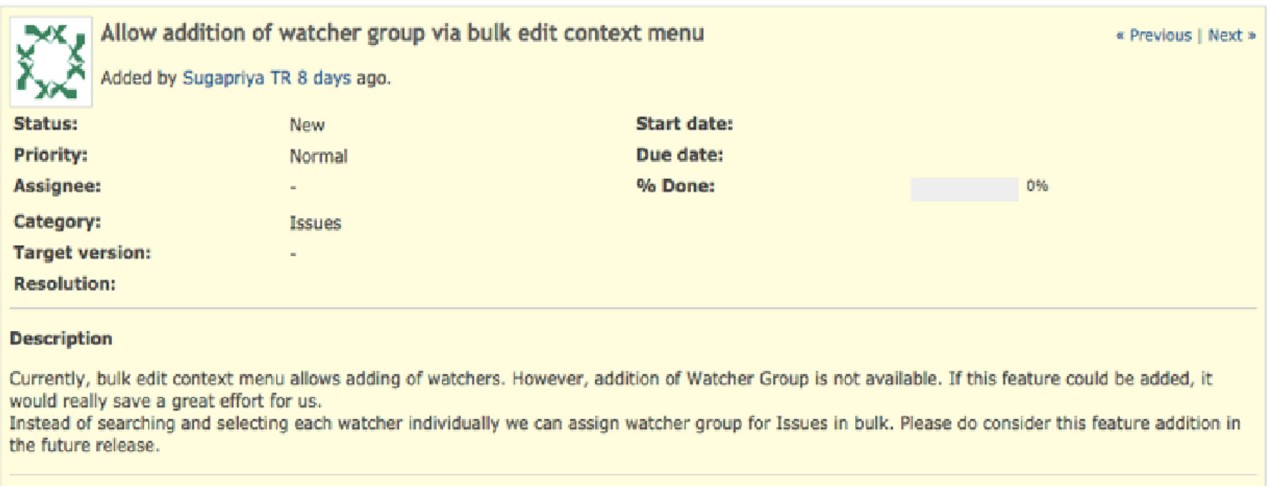

**Fig 4. Redmine project task without customized field.**

Our proposal does not solve the problem of infrastructure, or tool customization. Our aim is to study if the information of project tasks is enough and effective to reuse the curation of SO posts. The next Section presents the study performed regarding project tasks and their contexts.

## Project task context

We performed an investigation of the elements that might be related to project tasks, considering that such elements will be used by the implemented model to retrieve similarities between project tasks. Project Task Context is a set of elements that compose a project task. Project tasks are the project assets used in this study since the suggested association of a curated post to a project task will be based on the similarity of the project task. Investigating possible project task context elements is essential to help clarify what information from project tasks are available for similarity comparisons. In addition, this investigation is vital to avoid research bias.

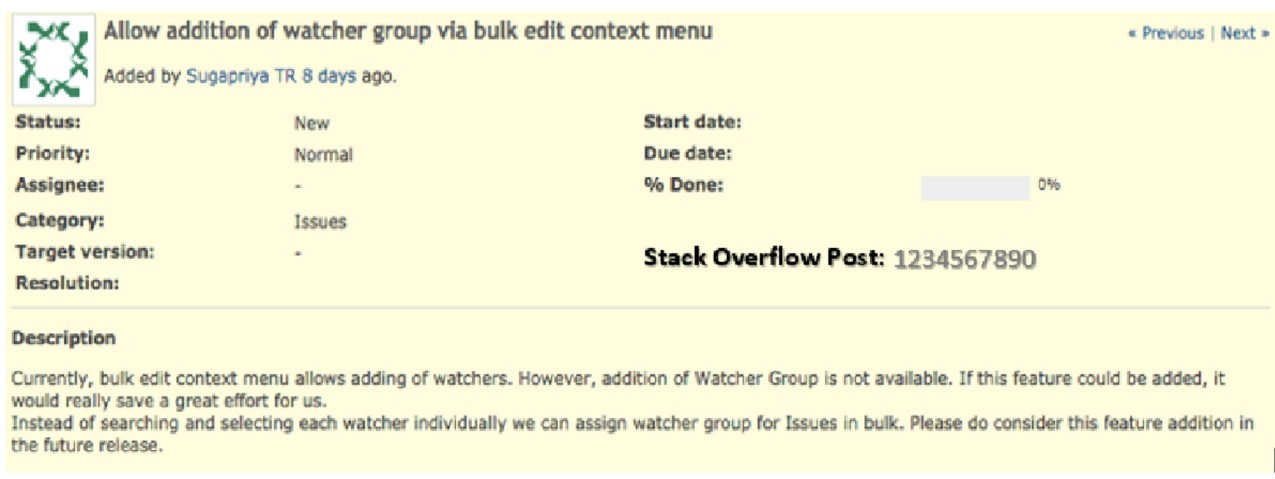

**Fig 5. Redmine project task with SO customized field—proposal.**

For completeness, we have taken advantage of both academic and industrial perspectives, including project management tools that support project tasks in software development.

We adopted two sources of information to determine the information present in a project task context. The first source was an **ad-hoc** literature review and the second was the information used to specify tasks in popular project management tools. From the literature, we found that software engineering is knowledge-intensive because of its dynamic nature and the amount of technology often encountered [1, 59]. According to Lindvall and colleagues [59], software engineering has both technical and business domain knowledge associated with the project. Technical knowledge refers to design (design patterns, heuristics, best practices, technical constraints, and estimation models), programming (programming languages and development tools) and software processes (methodology, code testing, and debugging procedures). Business domain knowledge refers to information regarding aspects of a specific application (business processes, business rules, activities, stakeholder needs, and business goals). This work does not consider business domain information; it only considers technical information because it aims to be agnostic to business characteristics. We propose a context element that captures technological information about project tasks such as a tag. A tag is a piece of information related to an element. In this case, elements are any technical information directly related to the task that can characterize the task. Each project has a specific context with respect to a product being developed [60], such as technical characteristics that every task will inherit necessarily, indicating the need for a project tag.

It is important to consider research sources other than just the formal literature in software engineering [61], to ensure that practical insights are brought to the activity. Therefore, we analyzed project management tools, by verifying the default elements each tool has for tasks. In this *ad-hoc* analysis, we concluded that some of the project task context elements identified in the literature were also identified in software development tools that support project workflows. The analyzed tools were JIRA (atlassian.com/Jira), Trello (trello.com) and Redmine (redmine.org), which are broadly used. After performing an analysis of both literature and project management tools contents, we present a list in Table 5 of the context elements that were identified.

## Similarity model implementation

After project task context elements were investigated in both the literature and project management tools, we proposed a method to find the similarities between project task context elements. The method comprises a process model that receives as input a dataset containing project tasks associated with SO posts, and retrieves similarity indices from pairs of project tasks. These indices evaluate if the SO posts are the same (an indication of potential reuse) between project tasks with a high degree of similarity. RapidMiner (RapidMiner Studio

**Table 5. Context elements.**

| Element | Description | Source |
|---|---|---|
| Project/Board | The name of the project that tasks belongs to | Redmine, Trello JIRA |
| Project Tag | Tags related to the project | Lindvall et al. [59] |
| Process | Process information that can be associated with the task | Lindvall et al. [59] |
| Title | The title of the task | Redmine, Trello JIRA |
| Description | The description of the task | Redmine, Trello JIRA |
| Category | A classification used to divide tasks into different niche | Redmine JIRA |
| Task Tag | Tags related to the task | Lindvall et al. [59], JIRA, Trello |

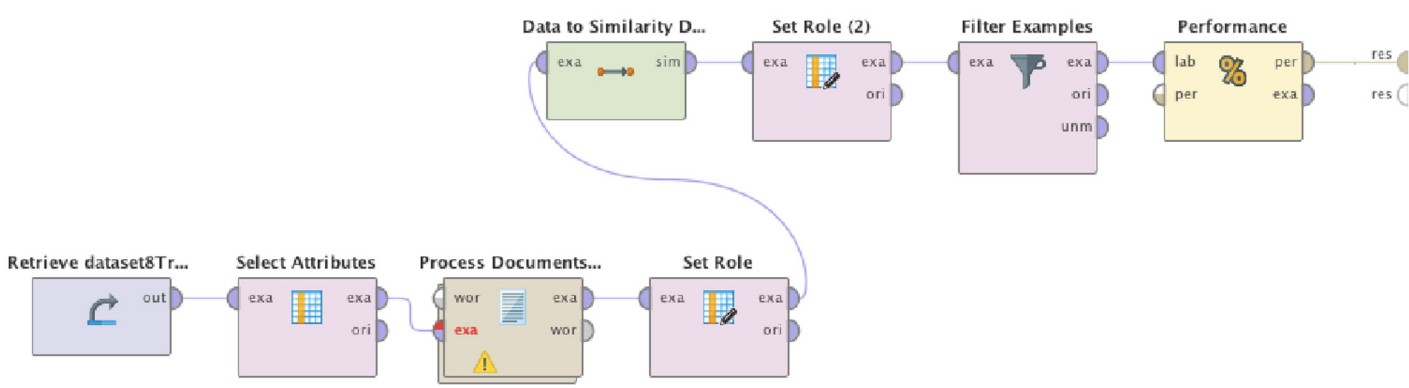

**Fig 6. RapidMiner process model.**

version 8.2) was used to implement a similarity process model. RapidMiner is a data science platform, requiring a small learning curve and widely adopted in the academic field [62].

RapidMiner implements a vast number of algorithms to extract text similarities. Each step of the RapidMiner process model is represented as an operator. In this sub-section, we present the implementation of the process model. The evaluation of the model is performed in Section Evaluation. The implemented RapidMiner process model is illustrated in Fig 6 and some details of the implemented model are described next.

In Fig 6, a straight line connects the sequential operators. Each operator runs a unique procedure, and the result is provided to the next operator. Operators have semicircles that are ports for inputs or outputs. Each operator has specific functionality. The **Retrieve Operator** imports a dataset. The **Select Attributes Operator** selects attributes from the dataset that are input to the similarity algorithm, to enable choosing a subset of elements from the dataset. The **Process Documents from Data Operator** is a sub-process responsible for the text pre-processing transformations. The transformations executed are the transformation of characters to lowercase (Transform Cases), the removal of every character that is not an alphanumeric character (Tokenize), stop-words filtering (Filter Stopwords) and finally, the transformation of inflected words into a base or root form of the word (Stem). This sub-process is presented in Fig 7.

This operator's output is connected to the next operator's input port, the **Set Role Operator** which is responsible for identifying the dependent variable. This dependent variable is needed by the **Data to Similarity Data Operator**, which is responsible for retrieving the text similarities, according to the algorithm chosen. The Data to Similarity Data operator is configured to run the Jaccard algorithm [63] at first, to retrieve text similarities, an algorithm that is widely used in this type of textual problem [64]. But there are other textual similarities that can be implemented in the model. Therefore, we have also used and we demonstrate results with different similarity algorithms. The similarities retrieved as a result of the execution of similarity algorithms have a range from 0 to 1 and can be interpreted in percentages (0 to 100%). The **Set**

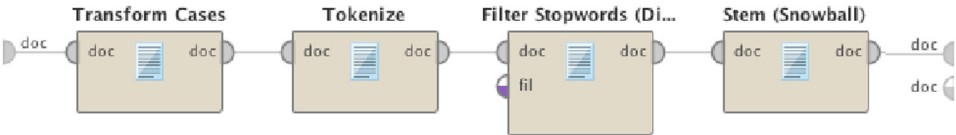

**Fig 7. Process document from data operator sub-process.**

**Role (2) Operator** identifies which values of the dataset will be chosen and which will be the base for comparison. The **Filter Examples Operator** sets a threshold of similarities. We defined a threshold of 50% similarity (similarity index $\geq$0.5). The threshold of 50% is chosen as this threshold is the mean between the minimum and maximum possible values, 0% and 100%, respectively. This threshold is not fixed, and can be changed depending on domain or design of application, or characteristics of the dataset used. It is defined as a way to eliminate false positives. The **Performance Operator** is used for statistical evaluation of classification tasks, and is configured to retrieve precision and accuracy metrics. Regarding the Jaccard similarity algorithm, a statistic metric used to compare finite sets of data is used to compare the similarity and diversity of sample sets and uses the ratio of the intersecting set to the union set as the measure of similarity. Thus, it equals to zero if there are no intersecting elements and equals to one if all elements intersect. An example of the equation of Jaccard similarity is presented below in the following equation.

$$JaccardSim(item1, item2) = \frac{|item1 \cap item2|}{|item1 \cup item2|}$$

Using documents as examples, supposing there are the three following documents:

- Doc1: "Word2", "Word3", "Word4", "Word2".

- Doc2: "Word1", "Word5", "Word4", "Word2".

- Doc3: "Word1"

Comparing Doc1 and Doc2, as the total number of unique words of both documents is 5, and the number of shared words between the documents is 2, gives a Jaccard similarity of 2/5 = 0.4. Comparing Doc1 and Doc1 gives a Jaccard similarity of 1. Comparing Doc1 and Doc3, the result is 0, because they share no similarities between the documents. As stated by Tan et al. [63], this is the most often used metric for document comparison.

## Evaluation

We evaluate the implemented model based on the guidelines proposed by Shani et al. [65]. The goal of this evaluation is to verify the effectiveness of using task similarities to reuse curated SO posts. The efficiency is conveyed through well-established precision and accuracy metrics. The research questions for the evaluation are:

**RQ1:** *What are the precision and the accuracy metrics for the collected sample?* It is essential to verify these metrics to gather quantitative results while ascertaining the effectiveness of considering similar project tasks to reuse curated SO posts. The metrics precision and accuracy were chosen after being the most common metrics identified in the Related Works.

**RQ2:** *What are the impacts in precision and accuracy when different context elements are combined?* It is essential to evaluate different project task context combinations because task contexts can vary in each project. A project can maintain records of processes while another project might not. Given this variation, it is important to understand the impacts of different project task context combinations.

**Hypothesis**: Project task similarity can provide helpful suggestions for curated SO posts. We test this hypothesis by verifying whether similar tasks (similarity above 50%) share the same SO posts.

**Controlling variables**: Considering this study uses only one dataset, having fixed controlled variables is not a concern. We propose a study considering different variable combinations to analyze the effects of the absence or presence of variables on precision and accuracy.

## Executing the implemented process model

To execute the process model, a dataset with project tasks has to be loaded into the RapidMiner process model implementation. The selected dataset was gathered from a company in Brazil that has been developing software products for more than 20 years and has 30 employees. The software development projects in this company follow agile guidelines, and the project tasks are managed with the support of a project management tool. We were able to gather 25 project tasks with associated SO posts for each of the 25 tasks. The software developers and managers of the company provided a spreadsheet with all information needed from tasks (context elements and SO posts associated with each task). The tasks required for this study should have two mandatory requirements: (i) all of the tasks should have at least one associated SO post and (ii) different tasks should share the same SO post. Our goal is to evaluate if different tasks having context similarities could share the same SO post. Tags were added to the dataset by the developers at the company. The company did not have tags previously associated with each project task but were able to identify tags for both the project and each project task for this study. With this dataset as input in the process model, we were able to execute the process model in RapidMiner. The dataset was composed by the following information regarding project tasks: Project, Category, Iteration, Title, Description, ProcessActivity, Project Tags, Task Tags and Stack Overflow Post. For a description of each of these elements, please refer to Table 5. The complete dataset and implementation can be found on github.com/glauciams/task2stackRapidMiner.

## Results and discussion

After executing the evaluation of the dataset, precision and accuracy are calculated, firstly using the Jaccard algorithm. Other similarity algorithms are also included and demonstrated in Table 8. A matrix is generated through the execution of the RapidMiner process model, which supports the extracted results.

Answering RQ1, the accuracy for the given dataset with the elements identified in Section Knowledge Reuse in Software Projects is 77.78%, and the precision mean 71.60%, when the threshold is set to 50% similarity. Table 6 presents the results reported in the Related Work section. The results presented are the highest scores reported in each of the related works, and the last row presents the findings from our current work. Depicting what these results represent, we argue that these numbers indicate a broader perspective of our investigation, namely the ability to reuse knowledge through the identification of project task similarities. Curated

**Table 6. RQ1 results and comparison with related works.**

| Related Work Articles | Precision | Accuracy |
|---|---|---|
| Ponzanelli et al., 2013 [5] | 2.43% [17] | 18.92% [17] |
| Rahman et al., 2014 [17] | 11% | 88% |
| Wu et al., 2018 [15] | 24.32% | |
| Correa et al., 2013 [14] | 47:27% | |
| Wang et al., 2015 [16] | 62% | |
| Current Work—RQ1 | 71.60% | 77.78% |

**Table 7. RQ2 results: Context combinations.**

| Precision | Accuracy | Combination changes |
|---|---|---|
| 77.78% | 85% | Removed Project and Project Tags |
| 71.67% | 54% | Removed Category, Process, Project, Project Tags and Task Tags |
| 71.60% | 77.78% | Current Work—RQ1 |
| 69.81% | 66.67% | Removed Project Tags and Task Tags |
| 61.46% | 74.47% | Removed ProjectTags |
| 61.46% | 74.47% | Removed Project |
| 61.17% | 70% | Included Interation |
| 54.76% | 38.28% | Removed TaskTags |
| 49.01% | 41.18% | Removed Category |
| 48.01% | 54.69% | Removed Process |
| 45.64% | 37.12% | Removed Title and Description |
| 45.64% | 37.12% | Removed Description |
| 40.87% | 36.57% | Removed Title |
| 13.51% | 22.69% | Removed Category, Process, Project, Title and Description |

Stack Overflow posts represent the tacit knowledge captured and reused in this proposal. For information purposes, we also disclose the results setting the algorithm similarity threshold to 0.3 (30%) and 0.7(70%) similarity. If the algorithm is set with a 0.3 threshold, the results are 40.87% for precision and 36.57% for accuracy. When set to 0.7, precision is 91.67% and accuracy is 94.44%.

Answering RQ2, Table 7 presents the context attributes' selected combination, the precision and accuracy extracted for each combination, and changes made in each combination, as it is not easy to perceive from the attribute list which attributes were selected and which were not in the combination. To facilitate the visualization of the information, we also provide a graph with the results, presented in Fig 8.

When considering the project task context elements initially identified in Section Knowledge Reuse in Software Projects, precision and accuracy are the highest among all combinations. When considering the Iteration context, which appeared in the dataset sample gathered in industry, precision and accuracy are the lowest, indicating that either the Iteration context element hampers SO posts reuse, or in the dataset sample, the information was not properly written. The same is perceived when removing other elements of the context, such as Process, ProjectTags and TaskTags, and Title and Description. These results also indicate that the hypothesis of the evaluation Project task similarity can provide useful suggestions of curated SO posts that are correct. There are indications that project task similarity can provide accurate associations between project tasks and curated SO posts, as the prediction and accuracy are as high as 70% and higher than most of the results reported in the related work. As well, the model built in RapidMiner can be used with other datasets with similar characteristics, which indicates the study is replicable with other data.

We have also run our model with different similarity algorithms, for comparison with Jaccard. The algorithms and results are demonstrated in Table 8. The threshold set for demonstration is 50%. It is not our goal to find the best algorithm or to study different algorithms in this study. Nevertheless, we decided to demonstrate the same model and dataset with different algorithms, to enrich our study and provide the means to compare Jaccard, one of the most used text similarities in our related work, to other available nominal and numeral distances and similarity algorithms. For the numeral distances (Euclidean, Cosine and Manhattan),

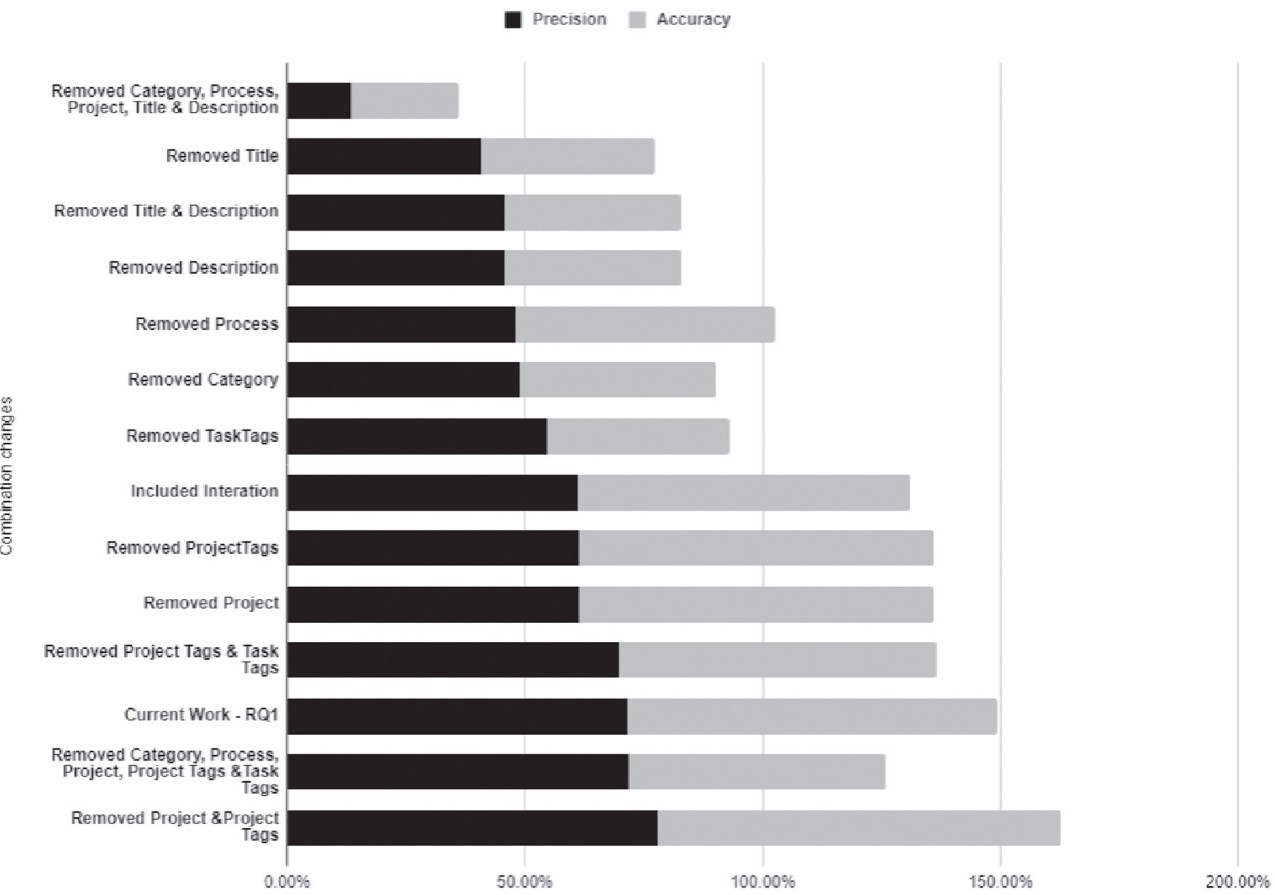

**Fig 8. RQ2 results: Context combinations in a graph.**

because there is no encoding, the title and description of the dataset are not considered, and we believe this is the reason why the values are low. Regarding the dataset sample characteristics, a higher number of tasks or different datasets might lead to a more robust general model to predict which SO Post to reuse when there are similar project tasks identified. Moreover, there has also been pointed out in the literature that researchers software development context that "... Although many research tools have been proposed that use historical information" (which is our case in this study, we use historical information) "..., few tools are available to practicing developers." [66]. Without tools that actually gather context from historical information, it is less likely there are available datasets for this matter. In conclusion, the sample size in the dataset is smaller than typically desired, yet the value of using real data seemed to outweigh the benefits of generating a simulated dataset.

**Table 8. Comparison of dataset with different similarity/distance algorithms.**

|  | Precision | Accuracy |
|---|---|---|
| Jaccard Similarity | 71.60% | 77.78% |
| Dice Similarity | 40.87% | 36.57% |
| Euclidean Distance | 8.28% | 15.71% |
| Cosine Similarity | 8.32% | 14.93% |
| Manhattan Distance | 3.88% | 12.98% |

With respect to the application of the proposed approach in other domains, we argue knowledge bases can be a useful asset in diverse domains such as language studies, health or mathematics. We believe these and other domains could use similar approaches to discover and reuse knowledge through automated tools and methods. Although the approach presented in our paper focuses on software development, the general principles of the proposed approach could be applied to knowledge bases in many additional domains. Of course, new approaches would require data from the experts using these knowledge bases. For example, in the case of health, physicians could take advantage of specific health information already curated by other experts that have used those knowledge bases. The approach could use clinical guideline tasks instead of software development tasks, and instead of Stack Overflow, the approach in health care could use a medical knowledge base Q&A website such as medhelp.org. The proposed approach could also be applied to knowledge bases in domains other than health such as mathematics (math.stackexchange.com) and the law (avvo.com).

According to the model proposed by Wohlin et al. [67], the **internal threats** to the validity of this evaluation are the small sample size, which can produce distortions in the expected result and conclusions. A strategy to mitigate this problem can include the use of more extensive samples. The lack of another sample and the small size of the dataset can also be characterized as **external threats** because it can impact the generalization of the study results. However, the selected dataset does not involve only one project, but considers multiple projects (five) of the company in an extended period of time (around 7 years). These multi-purpose projects are: (1) a legacy project with maintenance tasks, (2) a project that is new and uses modern programming languages for front-end, back-end and database—Ruby on Rails, NodeJS and MongoDB, (3) a main project in the company, that had in 2018 been in production for over 8 years and lastly, (4) a project created to coordinate the migration of the application server, and lastly, (5) a project that the system architects of the company managed their backlog. We consider this variability in the purpose of projects to be valuable to mitigate the reduced dataset threat. Moreover, although the sample size in the dataset is smaller than typically aspired, yet the importance of utilizing real data seemed to excel in the advantages of designing a simulated dataset. In the future, we plan to mitigate this threat of external validity by experimenting using a larger dataset. As **construction threats**, we can cite the selection of methods and definitions of measures, which were performed based on the related work, but gave no indication of being representative of the situation. Finally, the fact that the same developers can be involved in the study might indicate **conclusion threats**, given the possibility the text of project tasks can be standardized, in case they were written by the same person.

## Conclusions

Our work focused on associating curated SO posts to software development tasks. To pursue such goal we have first conducted a detailed SMS that allowed us to discover the strategies used to associate Software Development with SO (RQ1); what information such approaches have used (RQ2); what evaluation strategies were used (RQ3); and what results were reported by these strategies (RQ4). Our findings from this SMS showed no work proposed reusing curated SO posts. Then, we defined and implemented a process model using RapidMiner to assess if capturing and reusing curated SO posts was doable. Using this model, we have assessed and evaluated our hypothesis, compatible with related work. Results confirm that when a software developer is performing a task, and this task is similar to another task that has been associated with a SO post, the same post can be recommended to and reused by the developer. We believe that this approach can significantly advance the state of the art of software knowledge reuse by paving the way to support novel knowledge-project associations.

One limitation of this work is the sample size used in the evaluation. Although small, we also identified small samples in our related work, when evaluations were performed on data-sets with similar characteristics (data from real software development projects). As a consequence, results can be regarded as indications rather than generalizations. Regarding project tasks in the sample, there is a concern about how project tasks are created, who writes the text of tasks, and the completeness of the task. The company that provided data for this study maintains strong quality standards regarding tasks because their clients have full access to the project management tool. Moreover, although the company is not officially certified by an institute that guarantees the level of maturity of processes, it is strict concerning the mainte-nance of descriptions of artifacts, and emphasizing verbal and written communication of artifacts.

Future work might involve the development of a recommendation tool using the strategy proposed. A tool would allow the incorporation of rating mechanisms for given suggestions. The tool was not developed in this research, as we intended to focus the study on the possibility of the reuse of curated SO posts through task similarity. A deeper understanding of the role of project task context elements should be pursued, in a broader perspective, allowing the defini-tion of weights for specific contexts.

## Supporting information

**S1 Appendix. Systematic mapping study resulting catalog.**
(PDF)

## Acknowledgments

The authors thank the Computer Science Library Liaison Rebecca Hutchinson for the support during the Systematic Mapping Study.

## Author Contributions

**Conceptualization:** Glaucia Melo, Toacy Oliveira.

**Data curation:** Glaucia Melo.

**Formal analysis:** Glaucia Melo.

**Funding acquisition:** Paulo Alencar, Donald Cowan.

**Investigation:** Glaucia Melo.

**Methodology:** Glaucia Melo.

**Project administration:** Toacy Oliveira, Paulo Alencar, Donald Cowan.

**Resources:** Toacy Oliveira, Paulo Alencar, Donald Cowan.

**Software:** Glaucia Melo.

**Supervision:** Glaucia Melo, Toacy Oliveira, Paulo Alencar.

**Validation:** Glaucia Melo.

**Visualization:** Glaucia Melo.

**Writing – original draft:** Glaucia Melo.

**Writing – review & editing:** Glaucia Melo, Toacy Oliveira, Paulo Alencar, Donald Cowan.

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
