## [Decision Letter · Decision Letter 0]

6 Oct 2020

PONE-D-20-27912

Knowledge Reuse in Software Projects: Retrieving Software Development Q&A Posts Based on Project Task Similarity

PLOS ONE

Dear Dr. Melo,

Thank you for submitting your manuscript to PLOS ONE. After careful consideration, we feel that it has merit but does not fully meet PLOS ONE’s publication criteria as it currently stands. Therefore, we invite you to submit a revised version of the manuscript that addresses the points raised during the review process.

ACADEMIC EDITOR:

We look forward to receiving your revised manuscript.

Kind regards,

Talib Al-Ameri, Ph.D

Academic Editor

PLOS ONE

Journal Requirements:

2. We noted in your submission details that a portion of your manuscript may have been presented or published elsewhere.

[Main results of precision and accuracy are published in Melo, G., Oliveira, T., Alencar, P., & Cowan, D. (2019). Retrieving curated stack overflow posts from project task similarities. In International Conference on Software Engineering Knowledge Engineering (pp. 415-418).]

Reviewers' comments:

Reviewer's Responses to Questions

**Comments to the Author**

1. Is the manuscript technically sound, and do the data support the conclusions?

Reviewer #1: Yes

2. Has the statistical analysis been performed appropriately and rigorously? 

Reviewer #1: Yes

3. Have the authors made all data underlying the findings in their manuscript fully available?

Reviewer #1: Yes

4. Is the manuscript presented in an intelligible fashion and written in standard English?

Reviewer #1: Yes

5. Review Comments to the Author

Reviewer #1: This paper focused on software knowledge reuse on Stack Overflow based on project task similarity. Two research questions were conducted.

There are some issues that the authors might consider:

(1) The manuscript suffers from how to generalize the proposed approach to other datasets or other domains. This paper focused on a domain-specific problem. The article fails to provide insights and give generalized suggestions to more diversified readers in this regard.

(2) Some figures and tables are not clear and are not friendly for readability. The authors should consider using vector graphics such as eps and pdf.

(3) Some recently related works should be included.

LinkLive: discovering Web learning resources for developers from Q&A discussions

Learning to answer programming questions with software documentation through social context embedding

Leveraging Official Content and Social Context to Recommend Software Documentation

To Do or Not To Do: Distill crowdsourced negative caveats to augment api documentation

6. PLOS authors have the option to publish the peer review history of their article (what does this mean?). If published, this will include your full peer review and any attached files.

Reviewer #1: No

---

## [Author Response · Author response to Decision Letter 0]

19 Nov 2020

PONE-D-20-27912

Knowledge Reuse in Software Projects: Retrieving Software Development Q&A Posts Based on Project Task Similarity

PLOS One

In this document, we are addressing the reviewers' comments. We thank the reviewers for the additional comments and willingness to improve the quality of the publication with a very helpful review. We have made changes in the manuscript, addressing each of the reviewers’ comments.

Journal Requirements:

>Answer: We have reviewed the template specifications. 

2. We noted in your submission details that a portion of your manuscript may have been presented or published elsewhere.

[Main results of precision and accuracy are published in Melo, G., Oliveira, T., Alencar, P., & Cowan, D. (2019). Retrieving curated stack overflow posts from project task similarities. In International Conference on Software Engineering Knowledge Engineering (pp. 415-418).]

>Answer: (This answer is also in the Cover Letter as requested). A preliminary and shorter version (4 pages) of this paper was peer-reviewed and formally published in the International Conference on Software Engineering Knowledge Engineering (SEKE 2019). This previous paper was extended to 35 pages in several ways. First, the process model description is extensively detailed to support an exact and accurate replication of the proposed model. Second, a completely new and unpublished systematic mapping study was introduced. This systematic mapping study answers research questions regarding current proposals that associate Stack Overflow with the development environment. Third, the experimental studies have been extended by the additional distance and similarity algorithm calculations. Also, the paper has been significantly enhanced by extensions that provide additional details about the background, related work, case studies, and the analysis of the results. Given that the first publication is published as a short paper, we believe that the findings presented in our paper will appeal to the PLOS One Readers and academic community who subscribe to PLOS One. Our findings will allow your readers to accurately reproduce our proposed model, as we have included details about the implementation. Besides, the novel systematic mapping study advances the state-of-the-art by providing integrated information regarding current proposals that associate Stack Overflow and software development.

Reviewer #1: This paper focused on software knowledge reuse on Stack Overflow based on project task similarity. Two research questions were conducted.

There are some issues that the authors might consider:

(1) The manuscript suffers from how to generalize the proposed approach to other datasets or other domains. This paper focused on a domain-specific problem. The article fails to provide insights and give generalized suggestions to more diversified readers in this regard.

>Answer: 

We have added the following paragraph to the manuscript, in the Discussion Section. 

“With respect to the application of the proposed approach in other domains, we argue knowledge bases can be a useful asset in diverse domains such as language studies, health or mathematics. We believe these and other domains could use similar approaches to discover and reuse knowledge through automated tools and methods. Although the approach presented in our paper focuses on software development, the general principles of the proposed approach could be applied to knowledge bases in many additional domains. Of course, new approaches would require data from the experts using these knowledge bases. For example, in the case of health, physicians could take advantage of specific health information already curated by other experts that have used those knowledge bases. The approach could use clinical guideline tasks instead of software development tasks, and instead of Stack Overflow, the approach in health care could use a medical knowledge base Q&A website such as medhelp.org. The proposed approach could also be applied to knowledge bases in domains other than health such as mathematics math.stackexchange.com and the law avvo.com.”

Regarding our dataset, although we have relied on one dataset in the software engineering domain, the dataset is diverse and considers multiple projects (five) of the company in an extended period of time (around 7 years), not only one project. The projects in the dataset have different characteristics, such as (1) legacy project with few maintenance tasks, (2) project that is new and uses modern programming languages for front-end, back-end and database - Ruby on Rails, NodeJS and MongoDB, (3) main project in the company, that had in 2018 been in production for over 8 years and lastly, a (4) project created to coordinate the migration of the application server. We have added this project characterization to the manuscript. This dataset, which considers multiple projects, covers a diversity of cases in the software engineering domain. 

(2) Some figures and tables are not clear and are not friendly for readability. The authors should consider using vector graphics such as eps and pdf.

>Answer: We have converted all the figures to .eps for improved quality. 

(3) Some recently related works should be included.

LinkLive: discovering Web learning resources for developers from Q&A discussions

Learning to answer programming questions with software documentation through social context embedding

Leveraging Official Content and Social Context to Recommend Software Documentation

To Do or Not To Do: Distill crowdsourced negative caveats to augment api documentation

>Answer: We have added the recommended references in the Related Work Section.

---

## [Decision Letter · Decision Letter 1]

30 Nov 2020

Knowledge Reuse in Software Projects: Retrieving Software Development Q&A Posts Based on Project Task Similarity

PONE-D-20-27912R1

Dear Dr. Melo,

We’re pleased to inform you that your manuscript has been judged scientifically suitable for publication and will be formally accepted for publication once it meets all outstanding technical requirements.

Kind regards,

Talib Al-Ameri, Ph.D

Academic Editor

PLOS ONE

Additional Editor Comments (optional):

Reviewers' comments:

Reviewer's Responses to Questions

**Comments to the Author**

1. If the authors have adequately addressed your comments raised in a previous round of review and you feel that this manuscript is now acceptable for publication, you may indicate that here to bypass the “Comments to the Author” section, enter your conflict of interest statement in the “Confidential to Editor” section, and submit your "Accept" recommendation.

Reviewer #1: All comments have been addressed

2. Is the manuscript technically sound, and do the data support the conclusions?

Reviewer #1: Yes

3. Has the statistical analysis been performed appropriately and rigorously? 

Reviewer #1: Yes

4. Have the authors made all data underlying the findings in their manuscript fully available?

Reviewer #1: Yes

5. Is the manuscript presented in an intelligible fashion and written in standard English?

Reviewer #1: Yes

6. Review Comments to the Author

Reviewer #1: (No Response)

7. PLOS authors have the option to publish the peer review history of their article (what does this mean?). If published, this will include your full peer review and any attached files.

Reviewer #1: No

---

## [Editor Report · Acceptance letter]

7 Dec 2020

PONE-D-20-27912R1 

Knowledge Reuse in Software Projects: Retrieving Software Development Q&A Posts Based on Project Task Similarity 

Dear Dr. Melo dos Santos:

I'm pleased to inform you that your manuscript has been deemed suitable for publication in PLOS ONE. Congratulations! Your manuscript is now with our production department. 

Kind regards, 

on behalf of

Dr. Talib Al-Ameri 

Academic Editor

PLOS ONE